# The burden of mortality due to injury in Cabo Verde, 2018

**Ngibo Mubeta Fernandes** [1]*, **Maria da Luz Lima Mendonça** [1], **Lara Ferrero Gomez** [2]

**1** National Public Health Institute, Praia, Cabo Verde, **2** Department of Natural, Life and Environmental Sciences, University Jean Piaget, Praia, Cabo Verde

☯ These authors contributed equally to this work.

* ngibo.fernandes@insp.gov.cv

**Data Availability Statement:** Data cannot be shared publicly because a data access was not included in the authorization by National Commission for Data Protection. Data are available from the National Directorate of Heath for researchers who meet the criteria for access to

## Abstract

External causes continue to be one of the main causes of mortality in the world and Cabo Verde is no exception. Economic evaluations can be used to demonstrate the disease burden of public health problems such as injuries and external causes and support prioritization of interventions aimed at improving the health of the population. The objective of this study was to estimate the indirect costs of premature mortality in 2018 due to injuries and other consequences of external causes in Cabo Verde. Years of potential life lost, years of potential productive life lost and human capital approach were used to estimate the burden and indirect costs of premature mortality. In 2018, 244 deaths were registered due to injury and other consequences of external causes. Males were responsible for 85.4% and 87.73% of years of potential life lost and years of potential productive life lost, respectively. The cost of productivity lost due to premature death caused by injury was 4,580,225.91 USD. The social and economic burden due to trauma was substantial. There is a need for more evidence on the burden of disease due to injuries and their consequences, to support the implementation of targeted multi-sectoral strategies and policies for the prevention, management, and reduction of costs due to injuries in Cabo Verde.

## Introduction

The world loses nearly five million people annually to trauma, poisoning and other external causes, which corresponds to 9% of global deaths [1]. Injuries continue to have significant weight on the global disease burden, and are among the main causes of morbidity and mortality in adolescents, young people and adults [2]. In addition, injury represent a considerable public health problem, especially in developing countries, which are responsible for almost 90% of the world's fatalities [3, 4]. Furthermore, the burden of injuries and external causes of morbidity and mortality has a considerable social and economic impact. For example, road accidents are among the main causes of death in young people and cause losses of up to 3% of the Gross Domestic Product (GDP) globally and up to 5% in low- and middle-income countries [5].

Despite being a considerable public health problem, injuries and their consequences were not prioritized on the world agenda [1, 6]. However, recognizing the magnitude of injuries and other consequences of external causes, goals have been set globally to reduce premature

confidential data. Data can be requested through the National Director of Health, email: angela. gomes@ms.gov.cv.

**Funding:** The authors received no specific funding for this work.

**Competing interests:** The authors have declared that no competing interests exist.

mortality rates from suicides and road accidents, and to strengthen prevention and treatment measures of important determinants of injuries and external causes by 2030 [5, 7].

Injuries and the consequences of external causes are a major burden of disease globally, moreover, they mainly occur in developing countries, where few studies evaluating their economic impact and burden of diseases have been carried out [6]. Economic analysis of the implementation of interventions to assess the results and costs of these interventions aimed at improving the health of the population are recommended [4, 8] and are used to demonstrate the disease burden in monetary terms [9–11].

Injuries continue to be one of the main causes of mortality and morbidity in the world, and Cabo Verde is no exception. In 2018, Cabo Verde recorded 2838 deaths, of which 244 were due to trauma, which represented a mortality rate of five (5) deaths per 1000 inhabitants and were the second leading cause of admissions in central hospitals in the country [12]. In addition, the country recognizes injuries and external causes as a public health problem that deserve the proper framework in the national health policy and a multisectoral approach to address this problem [13].

Cabo Verde is one of the Small Island Developing States (SIDS), and is located 450–500 km from the west coast of Africa and is characterized by scarce natural resources, recurrent droughts, with a relatively small market economy. Cabo Verde is a lower middle-income country with a tourism-based economy. Political stability and good governance have contributed to the economic success of the country [14]. The archipelago is made up of 10 islands and has a land surface of 4,033 square kilometers [15] and had an estimated population of 544,081 in 2018 [16].

The current study is the first carried out in Cabo Verde estimating the indirect costs of mortality and demonstrating the socioeconomic burden due to injuries and consequences of external causes.

## Materials and methods

### General approach

The study is a cross-sectional analysis, with descriptive analysis of the epidemiological profile of premature mortality due to injury and a partial economic evaluation of the costs of premature mortality, adopting the socioeconomic perspective. In the mortality data used, deaths from injuries were classified into two groups: external causes and injuries and consequences of external causes. For this study, and all deaths classified in both groups were considered.

### Source and data collection

The Integrated Surveillance and Response Service of the National Health Directorate of the Ministry of Health and Social Security (MSSS) made available mortality data for 2018. The data were anonymized, in order to guarantee the confidentiality of individual data. Deaths were filtered by age, sex, municipality and cause of death. All deaths classified with codes S00-T98 and V01-Y98 of the International Classification of Diseases and Related Health Problems, Tenth Revision (ICD-10) were extracted for analysis [17].

Descriptive data analysis was performed using Microsoft Excel 2016 and Statistical Package for Social Science (SPSS, version 22) and Mendeley Reference Management Software [18], was used to organize the bibliographic references.

### Estimation methods

Life expectancy and the human capital approach were used to estimate the burden and costs of premature mortality. The human capital approach was applied to estimate the value of

potential lost productivity due to mortality due to injury [19]. To estimate the disease burden of premature mortality, measures of years of potential life lost (YPLL), years of potential productive life lost (YPPLL) and cost of productivity lost CPL were used [9, 10, 19, 20]. These concepts estimate the average time a person would live if they did not die prematurely, taking into account the age limit (death/retirement) and the cause [9]. YPLL estimates were based on the average value of life expectancy at birth in 2018 [21].

The YPLL were determined using the formula:

$$YPLL = \sum_{i=I}^{I} a_i d_i \qquad (1)$$

Where: $a_i$ = (73 −i−0.5) the number of years left to the stipulated age limit, $d_i$ = the number of deaths in the age between i and i + 1.

YPLL rates were stratified by municipality, sex, age group and cause of death. Demographic projections [16] were used to calculate the YPLL rates.

$$Rate\ of\ YPLL = \sum_{i=I}^{I} a_i d_i \times \frac{10^n}{N} \qquad (2)$$

Where: N = number of people between 1 and 72 years of age in the population. Note: ai and di are already defined above.

To estimate the YPPLL, the age groups from 15 to 64 years old constituted the national workforce [22].

$$YPPLL_j = \sum_{j=1}^{J} d_j \times W_j \qquad (3)$$

Where: $d_j$ = average number of deaths for each age group $W_j$ = midpoint for retirement age in each age group = 65-$j$-0.5 and $j$ = are groups of 5 years for the entire productive population (15–64).

The YPPLL and the Gross Domestic Product (GDP) *per capita* of 3617.33 USD for the year 2018 [21], were used to estimate the CPL due to premature mortality using the following formula:

$$CPL = \sum_{j=1}^{J} (YPPLL) \times GDP\ per\ capita \qquad (4)$$

For this study it was assumed that the individual's productivity did not change until retirement [9]. The present values of future costs were calculated by applying a discount rate of 4.5% practiced by the Bank of Cabo Verde [23]. In addition, a univariate sensitivity analysis was performed considering alternative discount rates (3% and 6%) to determine how different discount rate values would affect present productivity costs [8, 24].

The present value was obtained using the following formula:

$$Present\ value = CPL_j \sum_{j=1}^{J} \frac{1}{(1+r)^t} \qquad (5)$$

Where: $CPL$ = future costs for each age group, $r$ = discount rate, $t$ = average point to reach retirement age for each age group and $j$ = are groups of 5 years for the entire productive population (15–64).

## Ethical and legal considerations

This study was approved by the National Ethics Committee for Health Research (*CNEPS*) through resolution n˚ 33/2020 of May 28, 2020 and the National Commission for Data Protection (*CNPD*) through dispatch n˚ 183/2020. Being a retrospective study, no informed consent was required.

## Results

### Mortality profile

Cabo Verde recorded 244 deaths from injuries and external causes in 2018. Males were responsible for 84.4% (206) of deaths and around 23.3% (48) of deaths were among individuals with elementary occupations. Overall, age groups with the highest proportion of deaths were those between 20 to 24, 25 to 29 and 35 to 39, which represent 34.4% (84) of the cases registered in 2018 S1A Table in S1 File. Overall, the principal causes of death, were due to traumatic head injuries (23.0%; 56) and intentional self-harm (22.5%; 55). The main causes of death among females were assaults (21.1%; 8), followed by head injuries (18.4%; 7) and intentionally self-inflicted injuries which accounted for 15.8% (6) of deaths. S2B Table in S1 File.

### Years of potential life lost

Of the 244 deaths reported due to trauma, 211 (86.5%) were in individuals aged between 1 and 72 years old and these accounted for 7222 YPLL and on average, 34.2 YPLL were lost per death. Males lost on average of 33.7 YPLL and females lost 37.6 YPLL per death. Overall, the age range from 20 to 39 years, accounted for 60.3% or 4357 YPLL, and the same age group in the male population, accounted for 63.3% or 3906 of the YPLL. For females, the age group 1 to 4 years contributed 20% (210 YPLL) of the total YPLL in this group, compared to 1.9% (140) for males. Approximately 14 years were lost per 1000 inhabitants between 1 and 72 in 2018. Males and females had rates of 23.6 and 4.2 YPLL per 1000 inhabitants, respectively Table 1.

**Table 1. Years of potential life lost, proportions, means and rates per 1000 inhabitants by sex and age group, Cabo Verde, 2018.**

| Age group | Male | | | Female | | | Total | | |
|---|---|---|---|---|---|---|---|---|---|
| | YPLL | % | YPLL rate | YPLL | % | YPLL rate | YPLL | % | YPLL rate |
| 1 to 4 | 140 | 1.9 | 6.5 | 210 | 2.9 | 10.3 | 350 | 4.8 | 8.4 |
| 5 to 9 | 262 | 3.6 | 10.4 | 66 | 0.9 | 2.7 | 328 | 4.5 | 6.6 |
| 10 to 14 | 242 | 3.4 | 9.4 | 0.0 | 0.0 | 0.0 | 242 | 3.4 | 4.7 |
| 15 to 19 | 111 | 1.5 | 4.6 | 111 | 1.5 | 4.7 | 222 | 3.1 | 4.6 |
| 20 to 24 | 1212 | 16.8 | 47.5 | 152 | 2.1 | 6.2 | 1364 | 18.9 | 27.3 |
| 25 to 29 | 865 | 12.0 | 30.1 | 228 | 3.2 | 8.7 | 1092 | 15.1 | 19.9 |
| 30 to 34 | 729 | 10.1 | 27.1 | 0.0 | 0.0 | 0.0 | 729 | 10.1 | 14.5 |
| 35 to 39 | 1101 | 15.2 | 50.0 | 71 | 1.0 | 3.9 | 1172 | 16.2 | 29.0 |
| 40 to 44 | 610 | 8.4 | 36.7 | 31 | 0.4 | 2.1 | 641 | 8.9 | 20.5 |
| 45 to 49 | 230 | 3.2 | 17.1 | 77 | 1.1 | 6.0 | 306 | 4.2 | 11.7 |
| 50 to 54 | 246 | 3.4 | 20.4 | 21 | 0.3 | 1.6 | 267 | 3.7 | 10.7 |
| 55 to 59 | 279 | 3.9 | 29.5 | 47 | 0.6 | 4.3 | 326 | 4.5 | 16.1 |
| 60 to 64 | 105 | 1.5 | 18.4 | 42 | 0.6 | 5.1 | 147 | 2.0 | 10.5 |
| 65 to 69 | 33 | 0.5 | 10.2 | 0 | 0.0 | 0.0 | 33 | 0.5 | 4.0 |
| 70 to 72 | 6 | 0.1 | 5.2 | 0 | 0.0 | 0.0 | 6 | 0.1 | 2.1 |
| **Total** | **6170** | **85.4** | **23.6** | **1053** | **14.6** | **4.2** | **7222** | **100** | **14.0** |

### The geographic distribution of YPLL

The distribution of YPLL due to injury varied between regions. The municipality of Praia contributed to the highest proportion of YPLL with 27.4% (1978), followed by the island of São Vicente with 982 (13.6%) and Santa Catarina with 782 (10.8%). The municipality of Brava had the highest rate of 34.5 YPLL per 1000 inhabitants and the lowest rate was recorded in the municipality of São Lourenço dos Órgãos with 5.5 YPLL per 1000 inhabitants. The municipalities of São Domingos and Ribeira Grande Santiago revealed the highest averages of losses per death of 46.6 and 45.5 YPLL and victims died on average, at 26.4 and 27.5 years, respectively. The municipality of Santa Catarina de Fogo had the lowest mean per death of 18.8 YPLL and, subsequently, the highest mean per death, of 54.2 years. The mean age at death from external causes in 2018 was 38.8 years Table 2.

### Years of potential life lost by cause of death

Causes due to intentional self-harm contributed the largest proportion of the total YPLL (25.8%; 1862) and similarly, for males and was responsible for 26.6%(1639). The largest contributor of YPLL for females were deaths from assault and accounted for 29.4% (309) of total YPLL among females. Regarding the mean YPLL per death, deaths from toxic effects of substances of essentially non-medicinal origin had the highest mean of 53.5 YPLL for both sexes and the same group of causes had the highest mean among males, 45 YPLL/death while for females, the highest average was 70 YPLL per deaths in the groups "causes from other effects

**Table 2. Years of potential life lost, proportions, means and rates per 1000 inhabitants by municipalities, Cabo Verde, 2018.**

| Municipality | YPLL | % | YPLL rate | Mean | Mean age at death |
|---|---|---|---|---|---|
| Brava | 178 | 2.5 | 34.5 | 29.7 | 43.3 |
| Paul | 147 | 2.0 | 28.3 | 36.8 | 36.3 |
| Tarrafal de São Nicolau | 122 | 1.7 | 25.3 | 30.5 | 42.5 |
| Ribeira Brava | 152 | 2.1 | 24.0 | 38.0 | 35.0 |
| Porto Novo | 377 | 5.2 | 23.8 | 29.0 | 44.0 |
| Tarrafal | 346 | 4.8 | 20.3 | 34.6 | 38.4 |
| Santa Catarina | 782 | 10.8 | 17.9 | 34.0 | 39.0 |
| São Salvador do Mundo | 132 | 1.8 | 16.5 | 43.8 | 29.2 |
| Boavista | 248 | 3.4 | 14.5 | 41.3 | 31.8 |
| Ribeira Grande | 204 | 2.8 | 14.0 | 29.1 | 43.9 |
| São Domingos | 187 | 2.6 | 14.0 | 46.6 | 26.4 |
| São Miguel | 177 | 2.5 | 13.4 | 44.3 | 28.8 |
| Praia | 1978 | 27.4 | 12.7 | 42.1 | 30.9 |
| São Vincent | 982 | 13.6 | 12.5 | 30.7 | 42.3 |
| Ribeira Grande de Santiago | 91 | 1.3 | 11.4 | 45.5 | 27.5 |
| Santa Catarina do Fogo | 57 | 0.8 | 11.4 | 18.8 | 54.2 |
| Maio | 76 | 1.1 | 11.2 | 38.0 | 35.0 |
| São Felipe | 219 | 3 | 11.2 | 27.4 | 45.6 |
| Sal | 368 | 5.1 | 10.0 | 28.3 | 44.7 |
| Santa Cruz | 188 | 2.6 | 7.6 | 31.3 | 41.7 |
| Mosteiros | 66 | 0.9 | 7.5 | 33.0 | 40.0 |
| São Lourenço dos Órgãos | 36 | 0.5 | 5.5 | 35.5 | 37.5 |
| Others | 114 | 1.6 | - | 16.2 | 56.8 |
| **Cabo Verde** | **7222** | **100** | **14.0** | **34.2** | **38.8** |

**Table 3. Years of potential life lost, proportions and means by causes, Cabo Verde, 2018.**

| Causes of death | Male | | | Female | | | Total | | |
|---|---|---|---|---|---|---|---|---|---|
| | YPLL | % | Mean | YPLL | % | Mean | YPLL | % | Mean |
| Intentional self-harm | 1639 | 26.6 | 36 | 223 | 21.2 | 37 | 1862 | 25.8 | 36 |
| Injuries to the head | 1137 | 18.4 | 28 | 132 | 12.5 | 33 | 1268 | 17.6 | 28 |
| Injuries involving multiple body regions | 1056 | 17.1 | 36 | 168 | 15.9 | 34 | 1223 | 16.9 | 36 |
| Accidental drowning and submersion | 841 | 13.6 | 38 | 0 | 0.0 | 0 | 841 | 11.6 | 38 |
| Assault | 838 | 13.6 | 36 | 309 | 29.4 | 39 | 1147 | 15.9 | 37 |
| Certain early complications of trauma not classified elsewhere | 158 | 2.6 | 32 | 0 | 0.0 | 0 | 158 | 2.2 | 32 |
| Other and unspecified effects of external causes | 137 | 2.2 | 34 | 70 | 6.7 | 70 | 207 | 2.9 | 41 |
| Other accidental threats to breathing | 117 | 1.9 | 29 | 0 | 0.0 | 0 | 117 | 1.6 | 29 |
| Toxic effects of substances of chiefly non-medicinal origin | 91 | 1.5 | 45 | 70 | 6.7 | 70 | 161 | 2.2 | 54 |
| Burns and corrosions | 46 | 0.7 | 23 | 82 | 7.7 | 27 | 128 | 1.8 | 26 |
| Injuries to unspecified part of trunk, limb, or body region | 41 | 0.7 | 41 | 0 | 0.0 | 0 | 41 | 0.6 | 41 |
| Injuries to the abdomen, lower back, lumbar spine and pelvis | 36 | 0.6 | 36 | 0 | 0.0 | 0 | 36 | 0.5 | 36 |
| Injuries to the neck | 21 | 0.3 | 11 | 0 | 0.0 | 0 | 21 | 0.3 | 11 |
| Poisoning by drugs, medicaments and biological substances | 16 | 0.3 | 16 | 0 | 0.0 | 0 | 16 | 0.2 | 16 |
| Injuries to the hip and thigh | 0 | 0.0 | 0 | 0 | 0.0 | 0 | 0 | 0.0 | 0 |
| **Total** | **6170** | **100** | **34** | **1053** | **100** | **38** | **7222** | **100** | **34** |

and unspecified effects of external causes and toxic effects of substances of essentially non-medicinal origin" Table 3.

## Years of potential productive life lost

Overall, 187 deaths occurred in individuals aged between 15 to 64 years and resulted in 4768 YPPLL. Males and females, accounted for 87.7% (4183) and 12.3% (585) of total YPPLL, respectively. The highest proportions of YPPLL were observed in the age group 20 to 24 years and similarly for males and corresponded to 24.1% (1148), 24.4% (1020), respectively, while for females, the highest proportion of YPPLL was observed in the age group from 25 to 29 years (32.1%; 188). For both sexes, the YPPLL rate was 13.2 per 1000 inhabitants, 23 per 1000 inhabitants for males and 3 per 1000 inhabitants of the female population Table 4.

**Table 4. Years of potential productive life lost, proportions, means and rates per 1000 inhabitants by sex and age group, Cabo Verde, 2018.**

| Age group | Male | | | Female | | | Total | | |
|---|---|---|---|---|---|---|---|---|---|
| | YPPLL | % | YPPLL rate | YPPLL | % | YPPLL rate | YPPLL | % | YPPLL rate |
| 15 to 19 | 95 | 2.3 | 3.9 | 95 | 16.2 | 4.0 | 190 | 4 | 4.0 |
| 20 to 24 | 1020 | 24.4 | 40.0 | 128 | 21.9 | 5.2 | 1148 | 24.1 | 22.9 |
| 25 to 29 | 713 | 17.0 | 24.8 | 188 | 32.1 | 7.2 | 900 | 18.9 | 16.4 |
| 30 to 34 | 585 | 14.0 | 21.8 | 0 | 0.0 | 0.0 | 585 | 12.3 | 11.6 |
| 35 to 39 | 853 | 20.4 | 38.7 | 55 | 9.4 | 3.0 | 908 | 19 | 22.5 |
| 40 to 44 | 450 | 10.8 | 27.1 | 23 | 3.9 | 1.5 | 473 | 9.9 | 15.1 |
| 45 to 49 | 158 | 3.8 | 11.8 | 53 | 9.1 | 4.1 | 210 | 4.4 | 8.0 |
| 50 to 54 | 150 | 3.6 | 12.4 | 13 | 2.2 | 1.0 | 163 | 3.4 | 6.5 |
| 55 to 59 | 135 | 3.2 | 14.3 | 23 | 3.9 | 2.1 | 158 | 3.3 | 7.8 |
| 60 to 64 | 25 | 0.6 | 4.4 | 10 | 1.7 | 1.2 | 35 | 0.7 | 2.5 |
| **Total** | **4183** | **100** | **23.0** | **585** | **100** | **3.0** | **4768** | **100** | **13.2** |

**Table 5. Years of potential productive life lost, proportions, means and rates per 1000 inhabitants by municipalities, Cabo Verde, 2018.**

| Municipality | YPPLL | % | Rate | Mean | Mean age at death |
|---|---|---|---|---|---|
| Boavista | 138 | 2.9 | 10.9 | 27.5 | 38 |
| Brava | 130 | 2.7 | 37.9 | 21.7 | 43 |
| Maia | 8 | 0.2 | 1.6 | 7.5 | 58 |
| Mosteiros | 50 | 1.0 | 8.7 | 25.0 | 40 |
| Others | 58 | 1.2 | 0.2 | 8.2 | 57 |
| Paul | 63 | 1.3 | 17.6 | 20.8 | 44 |
| Porto Novo | 278 | 5.8 | 25.3 | 25.2 | 40 |
| Praia | 1380 | 28.9 | 12.5 | 32.9 | 32 |
| Ribeira Brava | 120 | 2.5 | 26.8 | 30.0 | 35 |
| Ribeira Grande | 148 | 3.1 | 14.7 | 21.1 | 44 |
| Ribeira Grande de Santiago | 75 | 1.6 | 13.8 | 37.5 | 28 |
| Sal | 273 | 5.7 | 10.5 | 24.8 | 40 |
| Santa Catarina | 485 | 10.2 | 15.9 | 24.3 | 41 |
| Santa Catarina do Fogo | 33 | 0.7 | 10.3 | 10.8 | 54 |
| Santa Cruz | 140 | 2.9 | 8.6 | 23.3 | 42 |
| São Domingos | 93 | 1.9 | 10.4 | 30.8 | 34 |
| São Filipe | 158 | 3.3 | 12.3 | 22.5 | 43 |
| São Lourenço dos Órgãos | 28 | 0.6 | 6.2 | 27.5 | 38 |
| São Miguel | 145 | 3.0 | 16.4 | 36.3 | 29 |
| São Salvador do Mundo | 50 | 1.0 | 9.4 | 25.0 | 40 |
| São Vicente | 615 | 12.9 | 10.7 | 22.0 | 43 |
| Tarrafal | 215 | 4.5 | 18.8 | 26.9 | 38 |
| Tarrafal de São Nicolau | 90 | 1.9 | 27.5 | 22.5 | 43 |
| **Cabo Verde** | **4768** | **100** | **13.2** | **25.5** | **40** |

### The geographic distribution of YPPLL

A total of 4768 YPPLL were calculated due to injuries and external causes, the municipality of Praia had the highest proportion of YPPLL (28.9%;1380), followed by the island of São Vicente (12.9%;615) and the municipality of Santa Catarina (10.2%;485). The geographic distribution of YPPLL rates was higher in the municipality of Brava with 37.9 per 1000 inhabitants, followed by the municipalities of the island of São Nicolau with 27.5 (Tarrafal) and 26.8 (Ribeira Brava). The municipalities of Ribeira Grande de Santiago and São Miguel recorded the highest losses, on average, of 37.5 and 36.3 YPPLL per death, respectively Table 5.

### Years of potential productive life lost by cause of death

Concerning the specific causes of death, for both sexes, intentional self-harm contributed 30.6% of YPPLL, followed by assaults with 19%, injuries involving multiple body regions with 17% and head injuries with 14%. On average, fatal victim of assaults recorded the highest mean YPPLL of 31, 32 and 31 for both sexes, for males and females, respectively. The overall mean YPPLL per death was 25, 26 YPPLL per death for males and 24 YPPLL for females. On average, males died at 39.3 years and females at 40.6 years Table 6.

### Estimated costs of productivity lost

The estimated cost of productivity lost due to premature death attributable to injuries and external causes was 4,580,225.91 USD. Males contributed 87.7% (4,053,833.79 USD) of the

**Table 6. Years of potential productive life lost, proportions and means by causes, Cabo Verde, 2018.**

| Causes of death | Male | | | Female | | | Total | | |
|---|---|---|---|---|---|---|---|---|---|
| | YPPLL | % | Mean | YPPLL | % | Mean | YPPLL | % | Mean |
| Intentional self-harm | 1283 | 30.7 | 30 | 175 | 29.9 | 29 | 1458 | 30.6 | 30 |
| Assault | 663 | 15.8 | 32 | 245 | 41.9 | 31 | 908 | 19.0 | 31 |
| Injuries involving multiple body regions | 725 | 17.3 | 28 | 70 | 12 | 18 | 795 | 16.7 | 27 |
| Injuries to the head | 655 | 15.7 | 19 | 38 | 6.4 | 13 | 693 | 14.5 | 19 |
| Accidental drowning and submersion | 435 | 10.4 | 24 | 0 | 0.0 | 0 | 435 | 9.1 | 24 |
| Certain early complications of trauma not classified elsewhere | 118 | 2.8 | 24 | 0 | 0.0 | 0 | 118 | 2.5 | 24 |
| Other and unspecified effects of external causes | 105 | 2.5 | 26 | 0 | 0.0 | 0 | 105 | 2.2 | 26 |
| Burns and Corrosions | 30 | 0.7 | 15 | 58 | 9.8 | 19 | 88 | 1.8 | 18 |
| Other accidental threats to breathing | 85 | 2.0 | 21 | 0 | 0.0 | 0 | 85 | 1.8 | 21 |
| injuries to unspecified part of trunk, limb, or body region | 33 | 0.8 | 33 | 0 | 0.0 | 0 | 33 | 0.7 | 33 |
| Injuries to the abdomen, lower back, lumbar spine and pelvis | 28 | 0.7 | 28 | 0 | 0.0 | 0 | 28 | 0.6 | 28 |
| Toxic effects of substances of chiefly non-medicinal origin | 13 | 0.3 | 13 | 0 | 0.0 | 0 | 13 | 0.3 | 13 |
| Poisoning by drugs, medicaments and biological substances | 8 | 0.2 | 8 | 0 | 0.0 | 0 | 8 | 0.2 | 8 |
| Injuries to the neck | 5 | 0.1 | 3 | 0 | 0.0 | 0 | 5 | 0.1 | 3 |
| Injuries to the hip and thigh | 0 | 0 | 0 | 0 | 0.0 | 0 | 0 | 0.0 | 0 |
| Total | 4183 | 100 | 26 | 585 | 100 | 24 | 4768 | 100 | 25 |

total CPL in 2018. The average CPL per death was 24,493.19 USD for both sexes, 24,870.15 USD for males and 21,933.01 USD for females. Approximately 21% (957,146.98 USD) of the total CPL was attributed to individuals between 35 to 39 years for both sexes, and 22%; (899,138.07 USD), for males, while for females, the highest proportion of CPL (24.2%; 127,341.62 USD) was observed in the 25–29 age group Table 7.

The principal drivers of CPL by specific causes were intentional self-harm, which was responsible for 27.5% (1,260,831.45 USD) of the total losses, followed by head injuries that contributed for 18.8% (862,712.50 USD). Furthermore, injuries to the abdomen, back, lumbar spine and pelvis had the highest mean loss per death, amounting to 29,004.45 USD. The same cause had the highest mean among males of 29,004.45 USD, while for females, the highest mean per death was attributed to the group intentionally self-inflicted injury in the amount of 25,374.22 USD followed by the cause "aggression" in the amount of 23,540.23 USD per death Table 7.

A univariate sensitivity analysis was conducted to assess the effects of varying the discount rates. The analysis, at 3% was worth the total CPL 6,900,400.84 USD and 3,140,890.80 USD at 6%, with an average CPP per death of 36,900.54 USD and 16,796.21 USD, respectively. At 3%, the highest proportion of the loss which corresponded to 20.8% (1,434,803.65 USD) of the total CPL, was observed in the 35 to 39 age group and the highest proportion of loss in the amount of 1,976,502.46 USD was attributed to the "intentional self-harm" group. At 6%, proportions of CPL by age group and cause did not differ significantly, and the total CPL value was approximately 50% lower than the total CPP value at 3% S3C and S4D Tables in S1 File.

## Discussion

This study demonstrated the socioeconomic burden due to injuries and external causes in 2018. The results of the study showed that mortality from injuries and consequences of external causes affected males disproportionately. Similar studies have shown significantly high mortality rates in males [25–27] and these inequalities between the sexes may be associated

**Table 7. Estimated costs of lost productivity by sex, age group and cause at 4.5%, Cabo Verde, 2018.**

| | Variables | Male | | | Female | | | Total | | |
|---|---|---|---|---|---|---|---|---|---|---|
| | | Present value | % | Mean | Present value | % | Mean | Present value | % | Mean |
| Age group | 15 to 19 | 41546.06 | 1.0 | 20773.03 | 41546.06 | 7.9 | 20773.03 | 83092.11 | 1.8 | 20773.03 |
| | 20 to 24 | 555888.66 | 13.7 | 23162.03 | 69486.08 | 13.2 | 23162.03 | 625374.74 | 13.7 | 23162.03 |
| | 25 to 29 | 483898.15 | 11.9 | 25468.32 | 127341.62 | 24.2 | 25468.32 | 611239.77 | 13.3 | 25468.32 |
| | 30 to 34 | 495115.37 | 12.2 | 27506.41 | 0.00 | 0.0 | 0.00 | 495115.37 | 10.8 | 27506.41 |
| | 35 to 39 | 899138.07 | 22.2 | 29004.45 | 58008.91 | 11.0 | 29004.45 | 957146.98 | 20.9 | 29004.45 |
| | 40 to 44 | 591460.80 | 14.6 | 29573.04 | 29573.04 | 5.6 | 29573.04 | 621033.84 | 13.6 | 29573.04 |
| | 45 to 49 | 257973.72 | 6.4 | 28663.75 | 85991.24 | 16.3 | 28663.75 | 343964.96 | 7.5 | 28663.75 |
| | 50 to 54 | 306173.51 | 7.6 | 25514.46 | 25514.46 | 4.8 | 25514.46 | 331687.97 | 7.2 | 25514.46 |
| | 55 to 59 | 343393.11 | 8.5 | 19077.39 | 57232.18 | 10.9 | 19077.39 | 400625.29 | 8.7 | 19077.39 |
| | 60 to 64 | 79246.35 | 2.0 | 7924.63 | 31698.54 | 6.0 | 7924.63 | 110944.89 | 2.4 | 7924.63 |
| | Total | **4053833.79** | **100** | **24870.15** | **526392.13** | **100** | **21933.01** | **4580225.91** | **100** | **24493.19** |
| Causes of death | Injuries to the head | 806706.02 | 19.9 | 23726.65 | 56006.48 | 10.6 | 18668.83 | 862712.50 | 18.8 | 23316.55 |
| | Injuries to the neck | 15849.27 | 0.4 | 7924.63 | 0.00 | 0.0 | 0.00 | 15849.27 | 0.3 | 7924.63 |
| | Injuries to the abdomen, lower back, lumbar spine and pelvis | 29004.45 | 0.7 | 29004.45 | 0.00 | 0.0 | 0.00 | 29004.45 | 0.6 | 29004.45 |
| | Injuries to the hip and thigh | 0.00 | 0.0 | 0.00 | 0.00 | 0.0 | 0.00 | 0.00 | 0.0 | 0.00 |
| | Injuries involving multiple body regions | 658778.30 | 16.3 | 25337.63 | 82043.39 | 15.6 | 20510.85 | 740821.69 | 16.2 | 24694.06 |
| | Injuries to unspecified part of trunk, limb, or body region | 27506.41 | 0.7 | 27506.41 | 0.00 | 0.0 | 0.00 | 27506.41 | 0.6 | 27506.41 |
| | Burns and corrosions | 48650.43 | 1.2 | 24325.22 | 47775.06 | 9.1 | 15925.02 | 96425.49 | 2.1 | 19285.10 |
| | Poisoning by drugs, medicaments and biological substances | 19077.39 | 0.5 | 19077.39 | 0.00 | 0.0 | 0.00 | 19077.39 | 0.4 | 19077.39 |
| | Toxic effects of substances of chiefly non-medicinal origin | 25514.46 | 0.6 | 25514.46 | 0.00 | 0.0 | 0.00 | 25514.46 | 0.6 | 25514.46 |
| | Other and unspecified effects of external causes | 101056.58 | 2.5 | 25264.15 | 0.00 | 0.0 | 0.00 | 101056.58 | 2.2 | 25264.15 |
| | Certain early complications of trauma not classified elsewhere | 126571.04 | 3.1 | 25314.21 | 0.00 | 0.0 | 0.00 | 126571.04 | 2.8 | 25314.21 |
| | Accidental drowning and submersion | 413219.68 | 10.2 | 22956.65 | 0.00 | 0.0 | 0.00 | 413219.68 | 9.0 | 22956.65 |
| | Other accidental threats to breathing | 112187.11 | 2.8 | 28046.78 | 0.00 | 0.0 | 0.00 | 112187.11 | 2.4 | 28046.78 |
| | Intentional self-harm | 1108586.12 | 27.3 | 25781.07 | 152245.33 | 28.9 | 25374.22 | 1260831.45 | 27.5 | 25731.25 |
| | Assault | 561126.51 | 13.8 | 26720.31 | 188321.86 | 35.8 | 23540.23 | 749448.37 | 16.4 | 25843.05 |
| | **Total** | **4053833.79** | **100** | **24870.15** | **526392.13** | **100** | **21933.01** | **4580225.91** | **100** | **24493.19** |

with risky behavior, the influence of society on men's behavior (masculinity) and in certain cases risky behavior is related to socioeconomic factors [28, 29]. In this study, the mean age of mortality due to trauma was 39 years and the most affected age groups were between 20 and 44 years. The findings of this study were similar to those reported in similar studies where the active population accounted for the largest proportion of fatal victims due to injury [26, 28].

The main causes of mortality by type of injury were traumatic head injuries followed by traumatic injuries involving multiple body regions. These results collaborate findings showing that head injuries were among the principal types of trauma that lead to death in patients [19, 30]. When considering only injuries with chapter XX codes, it was observed that self-harm, assault, and drowning were the predominate causes of death. In comparison Nobre *et al.* and Pillay-van Wyk *et al.* [31, 32] reported aggressions and traffic accidents as the main causes of death and while Malta *et al.* and Xing *et al.* [33, 34] reported traffic accidents and suicides as the principal external causes of death and finally, Corassa *et al.* [35] indicated to suicides and

aggressions as the most prevalent causes. The prevalence of suicide has been associated with mental disorders and sociodemographic characteristics and aggression with socioeconomic inequalities, alcohol and drug-related harm, respectively [26, 33]. In addition, traffic accidents were among the leading causes of death in Sub-Saharan Africa and while globally, traffic accidents, falls and interpersonal violence were identified as the leading causes of death due to injury [25, 36].

In this analysis, traffic accidents were not among the main causes of death in 2018. This was supported by data from other trauma surveillance systems that showed a low number of deaths due to traffic accidents [37]. Comprehensive road safety laws and their proper enforcement could contribute to the reduction of mortality from road accidents [25, 36]. Cabo Verde had approved a highway code in 2007 which has brought substantial gains in road safety [38].

In this study, assault was among the leading causes of death in the sample population and the principal cause among females. Cabo Verde has recognized gender-based violence as a social and public health problem and subsequently approved laws on the issue [39] and recognized that multiple socio-economic and cultural factors allow the existence of this phenomenon [40]. Furthermore, gender-based violence accounted for 20% of crimes reported [37] thus, supporting the results of the current study. Insufficiency in the implementation and regulation of laws could explain the prevalence of gender based violence [35]. Overall, mortality due to violence was prevalent in males than in females and has also been attributed to socioeconomic factors and other heath determinants [28, 29].

In the current analysis, drownings accounted for approximately 10% of mortality in the sample population, and 95.8% of these deaths were males. In Seychelles, drowning was the leading cause of death due to external causes [41]. The author associated this with the fishing industry, one of the country's main economic and leisure activities and the need to strengthen safety measures in these industries. Despite being a small island country, similar to Cabo Verde, Abio *et al*. [41] suggested the promotion of swimming programs as a measure to reduce the risk of drowning in the population. Drownings occurred mainly in adult males, however approximately 20% of deaths were in children in this analysis and drowning has been highlighted as one of the leading causes of death in children [35, 42].

The estimated YPLL corresponded to a total of 7,222 and, a mean of 34.2 YPLL per death, which was higher than the mean obtained in Tanzania [9] and in line with the findings from Brazil which showed that males were responsible for the highest proportion of calculated YPLL, while the age ranges from 20 to 39 were responsible for approximately 60% of YPLL [35]. The differences in the mean and prevalent age groups could explain the difference in life expectancy used to obtain YPLL and other socioeconomic factors. Similar to the current study, males contributed to a higher proportion of YPLL due to injuries, and overall injuries affected the potentially productive population [10, 35, 43]. Among the major contributors to YPLL were deaths due to self-harm, assault and drowning, this result corroborates findings by Delgado [44].

Deaths among individuals between 15 and 64 years of age resulted in 4768 YPPLL, on average 25 YPPLL per death. Intentional self-harm and aggression were responsible for approximately 50% of YPLL and YPPLL and suicides were responsible for 30% of YPPLL. Corassa *et al*. [35] and Rajabali *et al*. [45] demonstrated that a significant proportion of YPPLL was attributable to self-harm in their studies. Additionally, injuries and other consequences from external causes accounted for 10% of lost productivity in the WHO African region, with self-harm and interpersonal violence accounting for 87.2% of productivity losses due to intentional injuries [46]. In economic terms, YPLL and YPPLL in individuals aged 20 years presented the greatest socioeconomic loss because this age corresponds to the age at which the greatest

investment has been made by society and the beginning of the return on that investment to society [47].

In this study, the total cost of lost productivity due to premature death from injuries and external causes was 4,580,225.91 USD, men contributed 87.7% of the losses and overall each death resulted in losses amounting to 24,493.19 USD. In other studies, CPL due to injury was significantly higher among men than among women [10, 48]. Najafi *et al.* [10] reported that the highest average loss per death was due to trauma, amounting to 72,571 USD/death while Davey *et al.* [48] presented a mean value of approximately 2500 USD per death. A study conducted in a high-income country estimated values of lost productivity almost three times higher than the estimates found in this study [45]. Variations could be attributed to differences in the age limits, study periods, the GDP per capita and economic factors related to the countries in which the studies were conducted.

The present value of the CPL at 4.5% discount rate corresponded to approximately 0.2% of GDP in 2018, considering that GDP was estimated at 1.967 billion USD. This was lower than estimates of the economic burden due to road traffic accidents in low- and middle-income countries [5].

In the database used for this study, 47.1% (115) of the cases, deaths were classified by type of lesion, therefore the external causes that led to the appearance of the lesion were unknown, and 52.9% (129) were classified by external causes. ICD-10 guidelines recommended the use of codes from chapter XIX and XX to classify traumatic events and if only one code is used, chapter XX should be prioritized [17]. Results of this study demonstrate the need to standardize the classification of injuries and this could improve current estimates of mortality from external causes and complement national and global disease burden estimates [7, 26]. Standardization can be achieved through regular training of health professionals on current guidelines [41] and the digitization of health information systems.

Available data on economic costs of diseases, shows that two main methods are used to evaluate the economic impact, namely the friction method and the human capital method, with the human capital method being the most used methodology in cost-of-disease analyses [49]. It is argued that the friction method may result more accurate estimates of lost productivity losses when compared to estimates found using the human capital method [50]. For future studies, both methods maybe applied to better evaluate the economic impact of premature mortality.

## Limitations

Data analysis confirmed the incomplete classification of causes of death due to injury. This did not allow for analysis to be done exclusively by external causes or by the anatomical site of the lesion. The approach chosen made it possible to analyze all deaths recorded due to trauma, however it made it difficult to compare with other studies on the subject. This study demonstrated the need for standardization and training in the use of recommended tools for classifying injuries.

The methodology adopted did not include analysis of potential loses in individuals less than 1 and older than 72 years which could contribute to underestimation of the burden of injury. Analyzing the results presented in this study, the aforementioned limitations must be taken into account.

## Conclusion

The study demonstrated the socio-economic burden on the healthcare system and society due to injuries and showed that the costs of lost productivity due to premature mortality were

substantial. The analysis showed that males were disproportionately affected by injuries. In addition, the largest proportion of deaths due to injuries was in young people with the potential to contribute to the socio-economic development of the country. Attention was drawn to the prevalence and substantial monetary burden of self-harm, drowning and violence in the population sample. Finally, the results of this study could be used to assess the need for implementation of other public health policies concerning injuries, and targeted strategies for injury prevention.

## Supporting information

**S1 File.**
(RAR)

## Acknowledgments

Thanks goes to the Board of Directors of the National Public Health Institute and the General Directorate for Planning and Budgeting of the Ministry of Health, Cabo Verde for the support given in the process of carrying out this project and to the National Directorate of Health for authorizing the use of data.

## Author Contributions

**Conceptualization:** Ngibo Mubeta Fernandes.

**Data curation:** Ngibo Mubeta Fernandes.

**Formal analysis:** Ngibo Mubeta Fernandes.

**Investigation:** Ngibo Mubeta Fernandes, Maria da Luz Lima Mendonça, Lara Ferrero Gomez.

**Methodology:** Ngibo Mubeta Fernandes.

**Project administration:** Ngibo Mubeta Fernandes.

**Resources:** Maria da Luz Lima Mendonça, Lara Ferrero Gomez.

**Supervision:** Maria da Luz Lima Mendonça, Lara Ferrero Gomez.

**Validation:** Ngibo Mubeta Fernandes.

**Visualization:** Ngibo Mubeta Fernandes.

**Writing – original draft:** Ngibo Mubeta Fernandes, Maria da Luz Lima Mendonça, Lara Ferrero Gomez.

**Writing – review & editing:** Ngibo Mubeta Fernandes, Maria da Luz Lima Mendonça, Lara Ferrero Gomez.

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
