## [Decision Letter · Decision Letter 0]

19 Dec 2022

PONE-D-22-31820The burden of mortality due to injury in Cabo Verde, 2018PLOS ONE

Dear Dr. Fernandes,

Thank you for submitting your manuscript to PLOS ONE. After careful consideration, we feel that it has merit but does not fully meet PLOS ONE’s publication criteria as it currently stands. Therefore, we invite you to submit a revised version of the manuscript that addresses the points raised during the review process. The reviewers invited to evaluate this manuscript, assessed the submission carefully and provided major comments that are needed to be addressed to enhance the draft. Please revise the submission accordingly and resubmit the draft.

We look forward to receiving your revised manuscript.

Kind regards,

Sina Azadnajafabad, MD, MPH

Academic Editor

PLOS ONE

Journal Requirements:

Additional Editor Comments:

None.

Reviewers' comments:

Reviewer's Responses to Questions

**Comments to the Author**

1. Is the manuscript technically sound, and do the data support the conclusions?

Reviewer #1: Yes

Reviewer #2: Partly

Reviewer #3: Yes

Reviewer #4: Yes

2. Has the statistical analysis been performed appropriately and rigorously? 

Reviewer #1: Yes

Reviewer #2: Yes

Reviewer #3: I Don't Know

Reviewer #4: Yes

3. Have the authors made all data underlying the findings in their manuscript fully available?

Reviewer #1: Yes

Reviewer #2: Yes

Reviewer #3: No

Reviewer #4: Yes

4. Is the manuscript presented in an intelligible fashion and written in standard English?

Reviewer #1: Yes

Reviewer #2: Yes

Reviewer #3: No

Reviewer #4: Yes

5. Review Comments to the Author

Reviewer #1: This is a public health paper on burden and extent of premature death in Cobe Verde due to injuries. The paper is prepared well and here are some comments to strengthen the manuscript.

1- There are many grammatical shortcomings directed to this paper that need to be revised.

2- The format of citations are not consistent throughout the paper

3- Some abbreviations are not common in public health and are used without their wording and their definition

4- There are many calculations that could be transferred to appendix

5- Also there are many tables in results for many variables of interest that could be transferred to appendix or merged

6- Otherwise I believe it is an important topic and the manuscript is drafted well with beautiful discussion.

Reviewer #2: This is a well-written paper which I enjoyed reading. For the main part, it adopts standard methodologies to report on the burden of injury mortality in Cabo Verde. Analysis of the burden in lower and middle countries is still somewhat limited and this paper will contribute to the general literature as well as being of considerable interest to local policy makers.

The authors use the definition of injuries used by the global injury community – “all deaths classified with codes S00-T98 and V01-Y98 of the International Classification of Diseases and Related Health Problems, Tenth Revision (ICD-10) were extracted for analysis”.

Life expectancy and the human capital approach were used to estimate the burden and costs of premature mortality. The human capital approach was applied to estimate the value of potential lost productivity due to mortality due to injury.

When looking at the results, I was surprised by the very high percentage of male deaths and the absence of discussion about hip fracture mortality, which is generally quite high in most other jurisdictions. “The main causes of death among females were assaults (21.1%), followed by head injuries (18.4%) and intentionally self-inflicted injuries which accounted for 15.8% of deaths.

Delving deeper I realised that life expectancy at birth (2018) was used not life expectancy at each year of age as is the process of the Global Burden of Diseases study.

This will have the impact of underestimating the years of life lost as, for example, a person who has survived to age 35 will have a substantially greater life expectancy than a person aged 0. People who die in their 80s and 90s still lose YPLLs.

The focus also seems to be on working years of age, which would explain the absence of analysis on older people. Those who die > 72 years should contribute substantially to the burden of injury mortality given the high numbers.

The authors could mention all these limitations in their paper but if life expectancy data are available per year of age I would encourage them to follow the GBD mortality methodology and measure YLLs in the standard fashion.

I am happy with the other assumptions made: “For this study it was assumed that the individual's productivity did not change until retirement”; and “the present values of future costs were calculated by applying a discount rate of 4.5% practiced by the Bank of Cabo Verde” with alternative discount rates (3% and 6%) tested in a sensitivity analysis to determine how different discount rate values affect present productivity costs.

The section on limitations is quite good other than the mention of accepting 72 as an appropriate life expectancy for all ages.

“Data analysis confirmed the incomplete classification of causes of death due to injury. This did not allow for analysis to be done exclusively by external causes or by the anatomical site of the lesion. The approach chosen made it possible to analyze all deaths recorded due to trauma, however it made it difficult to compare with other studies on the subject. This study demonstrated the need for standardization and training in the use of recommended tools for classifying injuries. Analyzing the results presented in this study, the aforementioned limitations must be taken into account.”

Minor point

Line 62 – unnecessary ‘and’

Reviewer #3: I read thie manuscript carefully; this paper uses the data obtained from the Ministry of Health and Social Security of Cabo Verde to estimate Years of potential life lost and years of potential productive life lost. The subject and the results are interesting! However, while the data presented in the study are important and useful, and the methodology seems sound, for several reasons I believe it is not suitable for publication. Below I have provided some of my more major comments:

1. Since 2022 is nearly over, the data presented in the study should be updated as its last estimate is for about 5 years ago

2. The language needs significant improvements; many of the sentences and phrases are not suitable for an academic publication.

3. The manuscript presents many findings in tables while it does not have any figures. It is recommended to provide maps or bar charts for the measures.

4. While the sample size of deaths due to trauma is low (about 300) most of the data is provided in format of percentage while the low number of traumas might affect the proportions.

5. There are many typos such as missing spaces or et al. is not in Italic. These mistakes should be corrected

Reviewer #4: The burden of mortality due to injury in Cabo Verde, 2018

Interesting paper that fills a knowledge gap on this topic in the region and particularly in Cape Verde. The analysis of the data is interesting and brings real knowledge on the subject of injuries. However, I have some remarks that should be considered before publication.

- There are many statistical tables, I suggest that you try to keep some of them in the body of the text and refer the others to the appendices while making reference to them in the text.

- It would be nice to have a mapping of the data in the section The geographic distribution of YPLL. If you do not know how to make these choropleth maps, at least propose an actography of the Cape Verde archipelago.

- In the discussion I find that you do not discuss enough the question of head injuries which are the most important in your analyses.

- Similarly, you state that traffic accidents are not the main cause of death, however, in Cape Verde as elsewhere in Africa the production of data on road mortality is underestimated. Thank you for trying to qualify this statement.

- Finally, in general, it would be good to recall how mortality data are collected (sources) and produced according to the different fields of causes.

6. PLOS authors have the option to publish the peer review history of their article (what does this mean?). If published, this will include your full peer review and any attached files.

Reviewer #1: **Yes: **Esmaeil Mohammadi, MD MPH

Reviewer #2: No

Reviewer #3: No

Reviewer #4: No

---

## [Author Response · Author response to Decision Letter 0]

31 Jan 2023

Dear Editor,

Dear Reviewers,

We would like to thank to reviewers for their precious inputs. All of them have been addressed and the content of the manuscript has been improved accordingly. Please find below our point-by-point responses to the comments.

Reviewer #1: 

We thank the reviewer for their comments concerning our manuscript! We hope the changes make our manuscript more appealing.

1- There are many grammatical shortcomings directed to this paper that need to be revised.

Response: Thank you for the remark we have revised and made necessary corrections.

2- The format of citations are not consistent throughout the paper

Response: Thank you for your comment we have corrected and revised our manuscript.

3- Some abbreviations are not common in public health and are used without their wording and their definition

Response: Thank you for your comment we have corrected and revised some of the abbreviations in our manuscript. Definitions of the main concepts are presented in lines 82 to 84 and 104. 

4- There are many calculations that could be transferred to appendix

Response: Thank you for the remark we moved some of the calculations as suggested. Tables with data on all deaths due to injury were transferred to the appendix and also tables on sensitivity analysis at 3% and 6% were transferred. 

5- Also there are many tables in results for many variables of interest that could be transferred to appendix or merged.

Response: Thank you for the remark we moved some of the calculations/tables as suggested.

6- Otherwise I believe it is an important topic and the manuscript is drafted well with beautiful discussion.

Response: Thank you very much for your comment.

Reviewer #2: 

We thank the reviewer for their comments concerning our manuscript! We hope the changes make our manuscript more appealing.

This is a well-written paper which I enjoyed reading. For the main part, it adopts standard methodologies to report on the burden of injury mortality in Cabo Verde. Analysis of the burden in lower and middle countries is still somewhat limited and this paper will contribute to the general literature as well as being of considerable interest to local policy makers.

Response: Thank you for your remarks.

The authors use the definition of injuries used by the global injury community – “all deaths classified with codes S00-T98 and V01-Y98 of the International Classification of Diseases and Related Health Problems, Tenth Revision (ICD-10) were extracted for analysis”.

Response: Thank you for your comment.

Life expectancy and the human capital approach were used to estimate the burden and costs of premature mortality. The human capital approach was applied to estimate the value of potential lost productivity due to mortality due to injury.

Response: Thank you for your remarks.

When looking at the results, I was surprised by the very high percentage of male deaths and the absence of discussion about hip fracture mortality, which is generally quite high in most other jurisdictions. “The main causes of death among females were assaults (21.1%), followed by head injuries (18.4%) and intentionally self-inflicted injuries which accounted for 15.8% of deaths.

Response: Thank you for your remarks. Regarding this finding, when we analyzed data of the 38 deaths in the female population, the top three causes were assault, head injuries and intentional self-harm. Hip injuries came in at number 5. As we pointed out in the manuscript, we could not classify deaths exclusively by chapter XIX or XX of ICD 10 so this could have affected the results presented. We also observed that low numbers of hip injuries as a cause of death among males.

Delving deeper I realised that life expectancy at birth (2018) was used not life expectancy at each year of age as is the process of the Global Burden of Diseases study.

Response: Thank you for your remarks. We used the average life expectancy estimates available at the time and was based on methodology used by: Rumisha SF, George J, Bwana VM, Mboera LEG (2020) Years of potential life lost and productivity costs due to premature mortality from six priority diseases in Tanzania, 2006-2015. PLoS ONE 15(6): e0234300. https://doi.org/10.1371/ journal.pone.0234300. Furthermore, we do not have data available on life expectancy by age group.

This will have the impact of underestimating the years of life lost as, for example, a person who has survived to age 35 will have a substantially greater life expectancy than a person aged 0. People who die in their 80s and 90s still lose YPLLs.

Response: Thank you for your remarks. We used the average life expectancy estimates available at the time and was based on methodology used by: Rumisha SF, George J, Bwana VM, Mboera LEG (2020) Years of potential life lost and productivity costs due to premature mortality from six priority diseases in Tanzania, 2006-2015. PLoS ONE 15(6): e0234300. https://doi.org/10.1371/ journal.pone.0234300. Furthermore, we do not have data available on life expectancy by age group.

The focus also seems to be on working years of age, which would explain the absence of analysis on older people. Those who die > 72 years should contribute substantially to the burden of injury mortality given the high numbers.

Response: Thank you for your remarks. We used the average life expectancy estimates available at the time. In our methodology 72 years was the upper age limit. As the objective of this study was to estimate the burden and costs of premature mortality, we only considered individuals who died prematurely before the defined average life expectancy and retirement age, respectively.

The authors could mention all these limitations in their paper but if life expectancy data are available per year of age I would encourage them to follow the GBD mortality methodology and measure YLLs in the standard fashion.

Response: Thank you for your remarks. And we agree with you suggestion and will mention it as a limitation in our manuscript. For this study, we used calculations based on other studies such as Rumisha SF, George J, Bwana VM, Mboera LEG (2020) Years of potential life lost and productivity costs due to premature mortality from six priority diseases in Tanzania, 2006-2015. PLoS ONE 15(6): e0234300. https://doi.org/10.1371/ journal.pone.0234300. Furthermore, we do not have data available on life expectancy by age group or per year of age.

I am happy with the other assumptions made: “For this study it was assumed that the individual's productivity did not change until retirement”; and “the present values of future costs were calculated by applying a discount rate of 4.5% practiced by the Bank of Cabo Verde” with alternative discount rates (3% and 6%) tested in a sensitivity analysis to determine how different discount rate values affect present productivity costs.

Response: Thank you for your comment.

The section on limitations is quite good other than the mention of accepting 72 as an appropriate life expectancy for all ages.

Response: Thank you for your comment. We included a comment on life expectancy in our limitations.

“Data analysis confirmed the incomplete classification of causes of death due to injury. This did not allow for analysis to be done exclusively by external causes or by the anatomical site of the lesion. The approach chosen made it possible to analyze all deaths recorded due to trauma, however it made it difficult to compare with other studies on the subject. This study demonstrated the need for standardization and training in the use of recommended tools for classifying injuries. Analyzing the results presented in this study, the aforementioned limitations must be taken into account.”

Response: Thank you for your comment.

Minor point

Line 62 – unnecessary ‘and’

Response: Thank you for your comment.

Reviewer #3:

 I read thie manuscript carefully; this paper uses the data obtained from the Ministry of Health and Social Security of Cabo Verde to estimate Years of potential life lost and years of potential productive life lost. The subject and the results are interesting! However, while the data presented in the study are important and useful, and the methodology seems sound, for several reasons I believe it is not suitable for publication. Below I have provided some of my more major comments:

Response: We thank the reviewer for their comments concerning our manuscript! We hope the changes make our manuscript more appealing.

1. Since 2022 is nearly over, the data presented in the study should be updated as its last estimate is for about 5 years ago

Response: Thank you for your comment and we agree with your statement.

At the time of carrying out this study, the latest data available were from 2018.

2. The language needs significant improvements; many of the sentences and phrases are not suitable for an academic publication.

Response: Thank you for your comment. We attempted to use more suitable language as suggested.

3. The manuscript presents many findings in tables while it does not have any figures. It is recommended to provide maps or bar charts for the measures.

Response: We agree with your comment. We used tables which allowed us to summarize results with various variables into one item. We reduced the number of tables in the manuscript by transferring some of them to the appendix. 

4. While the sample size of deaths due to trauma is low (about 300) most of the data is provided in format of percentage while the low number of traumas might affect the proportions.

Response: Thank you for your comment and we agree with your comment about the sample size. Cabo Verde has a relatively small population and had a general mortality rate of 5.2 per 1000 population. We analyzed deaths reported due to injury for 2018 which resulted in a 211-sample size for YPLL and 187 sample size for YPPLL. When and where possible, we provided absolutes numbers, percentages, averages and rates. We also analyzed the sample size by sex to show the results within sexes and in the general population. 

5. There are many typos such as missing spaces or et al. is not in Italic. These mistakes should be corrected

Response: Thank you for your comment. We made an effort to correct typos and the language used. 

We followed reference guidelines provided by the journal. Guidelines on in-text citation of authors was not very clear to us (https://journals.plos.org/plosone/s/submission-guidelines#loc-references) however, other sources used did not require that et al. be in italics (https://guides.library.uwa.edu.au/vancouver). 

Reviewer #4:

 The burden of mortality due to injury in Cabo Verde, 2018

Interesting paper that fills a knowledge gap on this topic in the region and particularly in Cape Verde. The analysis of the data is interesting and brings real knowledge on the subject of injuries. However, I have some remarks that should be considered before publication.

Response: We thank the reviewer for their comments concerning our manuscript! We hope the changes make our manuscript more appealing.

- There are many statistical tables, I suggest that you try to keep some of them in the body of the text and refer the others to the appendices while making reference to them in the text.

Response: Thank you for your comment. We transferred some of the tables to the appendix.

- It would be nice to have a mapping of the data in the section The geographic distribution of YPLL. If you do not know how to make these choropleth maps, at least propose an actography of the Cape Verde archipelago.

Response: Thank you for your suggestion. We included maps of Cabo Verde with the distribution of the YPLL and YPPLL rates by municipality.

- In the discussion I find that you do not discuss enough the question of head injuries which are the most important in your analyses.

Response: Thank you for your comment. We cited literature discussing head trauma. Unfortunately, we worked with data from a database and did not have information on mechanisms that lead to the head traumas and we were limited our discussion on the topic. 

- Similarly, you state that traffic accidents are not the main cause of death, however, in Cape Verde as elsewhere in Africa the production of data on road mortality is underestimated. Thank you for trying to qualify this statement.

As we mentioned in our manuscript, Data on road accidents is also provided annually by the National Police Service (number of accidents, number of deaths…). Furthermore, completeness of death registration in Cabo Verde is over 90% and among the highest in Africa(https://www.niussp.org/environment-and-development/vital-registration-africa-will-completea-quand-un-etat-civil-exhaustif-en-afrique/). We reported discrepancies between the numbers reported by the police service and the ministry of health and believe that this could be due to incomplete classification of some deaths, for example some of the deaths reported by anatomic region could have been to road accidents… 

- Finally, in general, it would be good to recall how mortality data are collected (sources) and produced according to the different fields of causes.

Response: Thank you for your comment. We mentioned the source of the data in our manuscript “The Integrated Surveillance and Response Service of the National Health Directorate of the Ministry of Health and Social Security (MSSS) made available mortality data for 2018. The data were anonymized, in order to guarantee the confidentiality of individual data. Deaths were filtered by age, sex, municipality and cause of death. All deaths classified with codes S00-T98 and V01-Y98…”

Thanks for your suggestions and recommendations! They helped us improve the paper substantially.

Best regards

---

## [Decision Letter · Decision Letter 1]

20 Feb 2023

The burden of mortality due to injury in Cabo Verde, 2018

PONE-D-22-31820R1

Dear Dr. Fernandes,

We’re pleased to inform you that your manuscript has been judged scientifically suitable for publication and will be formally accepted for publication once it meets all outstanding technical requirements.

Kind regards,

Sina Azadnajafabad, MD, MPH

Academic Editor

PLOS ONE

Additional Editor Comments (optional):

Reviewers' comments:

Reviewer's Responses to Questions

**Comments to the Author**

1. If the authors have adequately addressed your comments raised in a previous round of review and you feel that this manuscript is now acceptable for publication, you may indicate that here to bypass the “Comments to the Author” section, enter your conflict of interest statement in the “Confidential to Editor” section, and submit your "Accept" recommendation.

Reviewer #1: All comments have been addressed

Reviewer #2: All comments have been addressed

Reviewer #3: All comments have been addressed

Reviewer #4: All comments have been addressed

2. Is the manuscript technically sound, and do the data support the conclusions?

Reviewer #1: Yes

Reviewer #2: Yes

Reviewer #3: Yes

Reviewer #4: No

3. Has the statistical analysis been performed appropriately and rigorously? 

Reviewer #1: Yes

Reviewer #2: Yes

Reviewer #3: Yes

Reviewer #4: Yes

4. Have the authors made all data underlying the findings in their manuscript fully available?

Reviewer #1: Yes

Reviewer #2: Yes

Reviewer #3: Yes

Reviewer #4: Yes

5. Is the manuscript presented in an intelligible fashion and written in standard English?

Reviewer #1: Yes

Reviewer #2: Yes

Reviewer #3: Yes

Reviewer #4: Yes

6. Review Comments to the Author

Reviewer #1: I believe authors have addressed all my comments and concerns and i have no more suggestions to enclose

Reviewer #2: I have no additional comments to make other than to perhaps reference the GBD methodology and state that you did not have the data to follow that methodology. The authors cannot do more with data they do not have.

Reviewer #3: (No Response)

Reviewer #4: I have no particular comment to make. All the remarks have been taken into consideration. The article can be published. If the authors have the possibility, a map could simplify the understanding by locating all the mentioned localities.

Thank you.

7. PLOS authors have the option to publish the peer review history of their article (what does this mean?). If published, this will include your full peer review and any attached files.

Reviewer #1: **Yes: **Esmaeil Mohammadi, MD MPH

Reviewer #2: No

Reviewer #3: No

Reviewer #4: No

---

## [Editor Report · Acceptance letter]

3 Mar 2023

PONE-D-22-31820R1 

The burden of mortality due to injury in Cabo Verde, 2018 

Dear Dr. Fernandes:

I'm pleased to inform you that your manuscript has been deemed suitable for publication in PLOS ONE. Congratulations! Your manuscript is now with our production department. 

Kind regards, 

on behalf of

Dr. Sina Azadnajafabad 

Academic Editor

PLOS ONE